# Gremlin-1 Promotes Colorectal Cancer Cell Metastasis by Activating ATF6 and Inhibiting ATF4 Pathways

**DOI:** 10.3390/cells11142136

**Published:** 2022-07-07

**Authors:** Ruohan Li, Huaixiang Zhou, Mingzhe Li, Qiuyan Mai, Zhang Fu, Youheng Jiang, Changxue Li, Yunfei Gao, Yunping Fan, Kaiming Wu, Clive Da Costa, Xia Sheng, Yulong He, Ningning Li

**Affiliations:** 1Tomas Lindahl Nobel Laureate Laboratory, The Seventh Affiliated Hospital of Sun Yat-sen University (SYSU), No. 628, Zhenyuan Road, Guangming Dist., Shenzhen 518107, China; lirh37@mail2.sysu.edu.cn (R.L.); maiqy6@mail2.sysu.edu.cn (Q.M.); fuzhang@mail2.sysu.edu.cn (Z.F.); jiangyh59@mail2.sysu.edu.cn (Y.J.); lichx59@mail2.sysu.edu.cn (C.L.); gaoyf25@mail.sysu.edu.cn (Y.G.); 2Center for Digestive Disease, The Seventh Affiliated Hospital of Sun Yat-sen University (SYSU), No. 628, Zhenyuan Road, Guangming Dist., Shenzhen 518107, China; limzhe5@mail.sysu.edu.cn (M.L.); wukaiming@sysush.com (K.W.); 3Guangdong Provincial Key Laboratory of Digestive Cancer Research, The Seventh Affiliated Hospital of Sun Yat-sen University, No. 628 Zhenyuan Road, Shenzhen 518107, China; 4Department of Otolaryngology, The Seventh Affiliated Hospital of Sun Yat-sen University, No. 628, Zhenyuan Road, Guangming Dist., Shenzhen 518107, China; fanyp@mail.sysu.edu.cn; 5The Francis Crick Institute, 1 Midland Road, London NW1 1AT, UK; clive.dacosta@crick.ac.uk; 6Ministry of Education Key Lab of Environment and Health, School of Public Health, Tongji Medical College, Huazhong University of Science and Technology, Wuhan 430074, China; 7China-UK Institute for Frontier Science, Shenzhen 518107, China

**Keywords:** Gremlin-1, epithelial–mesenchymal transition, ATF4, ATF6, colorectal cancer

## Abstract

Cancer cell survival, function and fate strongly depend on endoplasmic reticulum (ER) proteostasis. Although previous studies have implicated the ER stress signaling network in all stages of cancer development, its role in cancer metastasis remains to be elucidated. In this study, we investigated the role of Gremlin-1 (GREM1), a secreted protein, in the invasion and metastasis of colorectal cancer (CRC) cells in vitro and in vivo. Firstly, public datasets showed a positive correlation between high expression of GREM1 and a poor prognosis for CRC. Secondly, GREM1 enhanced motility and invasion of CRC cells by epithelial–mesenchymal transition (EMT). Thirdly, GREM1 upregulated expression of activating transcription factor 6 (ATF6) and downregulated that of ATF4, and modulation of the two key players of the unfolded protein response (UPR) was possibly through activation of PI3K/AKT/mTOR and antagonization of BMP2 signaling pathways, respectively. Taken together, our results demonstrate that GREM1 is an invasion-promoting factor via regulation of ATF6 and ATF4 expression in CRC cells, suggesting GREM1 may be a potential pharmacological target for colorectal cancer treatment.

## 1. Introduction

Colorectal cancer (CRC) is the third most common cancer worldwide and the second leading cause of cancer-related deaths [1]. CRC has accounted for 10% of malignancies and its mortality rate remains categorically high. More than half of CRC patients have metastases at the time they are diagnosed [2]. The current standard of care for CRC includes surgery, chemotherapy, radiation therapy and targeted therapy, with a 5-year relative survival rate of 65% for CRC patients, which devastatingly declines to 12% in patients with stage IV CRC [3].

Gremlin-1 (GREM1), a secreted glycoprotein, belongs to the DAN/Cerberus protein family, which is a member of the cysteine knot superfamily that includes, among others, transforming growth factor-β (TGF-β) and vascular endothelial growth factor (VEGF) [4]. It is also a BMP antagonist and known to be involved in embryogenesis [4,5], bone formation [6] and organ development [7]. In pathological states, GREM1 is involved in processes such as organ fibrosis [8], inflammation [9] and cancer [10,11]. Above all, GREM1 is known to induce fibrosis of organs, which requires the epithelial–mesenchymal transition (EMT) process [12,13]. Cancer-associated fibroblasts (CAFs) express GREM1, which inhibits BMP signaling and accelerates tumor cell proliferation [14]. In patients with breast cancer or CRC, high expression of GREM1 is usually associated with poor prognosis [15,16,17]. GREM1 secreted by glioma cancer stem cells maintains its own stemness and proliferation, whilst blocking differentiation of glioma cells [18]. Overexpression of GREM1 in intestinal epithelial cells supports the colorectal premalignant lesions [11,19]. In contrast, a clinical study showed that high expression of GREM1 was correlated with good prognosis in CRC [20]. Thus, the functional role of GREM1 in CRC remains elusive.

The endoplasmic reticulum (ER) in eukaryotic cells is involved in a variety of functions comprising lipid biosynthesis, calcium storage, protein transport and protein folding [21,22]. Stress stimuli disturb ER proteostasis [23,24], which leads to the activation of the unfolded protein response (UPR). The UPR signaling network is initiated by three ER transmembrane sensors, IRE1α, PERK and ATF6, and functions either to restore ER proteostasis or to confer apoptosis depending on the context [25,26,27]. In recent years, ER stress and UPR signaling have been extensively studied in various types of cancers, including both tumor cells per se, and the stromal cells within their microenvironment [28,29,30]. In the light of these efforts, targeting the key nodes of the UPR network has been proposed for novel therapeutic strategies in cancer treatment, and is under rapid translational development [31,32]. Despite this positive progress, relatively less is known about the implication of ER stress signaling in cancer metastasis. A recent study revealed that EMT was triggered by activation of the IRE1a and PERK pathways in lung adenocarcinoma cells A549 and H358 [33]. In pancreatic adenocarcinoma cells Capan-2 and SW1990, intracellular Ca^2+^ overload also triggered EMT through the IRE1a pathway [34]. However, its relevance to CRC metastasis has not been studied.

In this study, we performed immunohistochemistry (IHC) of GREM1 in CRC clinical specimens, and found that its expression was associated with poor prognosis. GREM1 activated EMT in CRC cells, which was mediated by the upregulation of ATF6 and downregulation of ATF4 pathways of the UPR. Further, our study preliminarily showed that GREM1 modulated the expression of ATF4 and ATF6 through BMP and VEGF signaling pathways, respectively. These results suggest that the crosstalk between GREM1 and the ER stress signaling network may provide a new theoretical basis for the treatment of advanced CRC.

## 2. Materials and Methods

### 2.1. Cell Lines and Culture

Human CRC cell lines (SW480 and HCT116), requested from the “Chinese Academy of Sciences”, were confirmed fingerprinting using “STR DNA” by Shanghai Biowing Applied Biotechnology Co., Ltd. (Shanghai, China). A Mycoplasma Stain Assay Kit (Beyotime, C0296) was used to identify the mycoplasma infection. The HCT116 cell line was cultivated with RPMI 1640 (Thermo Fisher Scientific, 31870082, Fremont, CA, USA) with 10% fetal bovine serum (FBS, Thermo Fisher Scientific), 100 μg/mL streptomycin and 100 U/mL penicillin (Strep/Pen, Thermo Fisher Scientific, 15140148) under the conditions of 5% CO_2_ at 37 °C. Similarly, SW480 cell line was cultivated with Dulbecco’s modified Eagle’s medium (DMEM, Thermo Fisher Scientific, 11965092) containing 10% FBS, 100 μg/mL streptomycin and 100 U/mL penicillin (Strep/Pen, Thermo Fisher Scientific, 15140148) in a humidified, 5% CO_2_ incubator at 37 °C.

### 2.2. Lentivirus Production and Infection

The shRNA was in the PTSB-SH-copGFP-2A-PURO lentiviral vector (shGREM1). The *GREM1* gene was ligated into a pLV-EF1a-IRES-PURO lentiviral vector to construct the plasmid for *GREM1* overexpression. To produce lentiviral particles, 1 × 10^7^ HEK293T cells in a 55 cm^2^ dish were co-transfected with 10 μg lentiviral vector, 5 μg pCMV-dR8.91 (packaging-expressing plasmid) and 5 μg pMD2.G (envelope-expressing plasmid). The supernatant containing viral particles was harvested at 48 and 72 h post transfection, and was filtered through a Millex-GP Filter Unit (0.45 μm pore size, Millipore, SLHV033RB). To infect cancer cells with lentivirus, cells were infected twice at a 24 h interval with culture medium containing 2 mL lentivirus, 200 μL FBS and 5 mg/mL polybrene (Sigma, TR-1003, St. Louis, MO, USA) at 37 °C. To increase the overexpression and knockdown efficiency, infected cells underwent several days of puromycin (Sigma, P8833) selection. For lentiviral infection, a *GREM1* expression vector (pLV-GREM1), a pLV-EF1a-IRES-PURO control lentiviral vector (pLV), the shRNA targeting *GREM1* (shGREM1) and a PTSB-SH-copGFP-2A-PURO lentiviral vector (shNC) were designed and constructed by TranSheepBio Medical Biotechnology Co., Ltd. (Shanghai, China) The sequence of shRNA oligo-nucleotides against *GREM1* (shGREM1) was: 5′-CCGGGCAGTGTCGTT-GCATATCCATCTCGAGATGGATATGCAACGACACTGCTTTTTT-3′. The mRNA and protein expression levels of GREM1 were then analyzed by qRT-PCR and Western blotting, respectively.

### 2.3. RNA-Seq and Gene Set Enrichment Analysis (GSEA)

Total RNA was isolated using the Trizol reagent (Invitrogen, 15596026, Carlsbad, CA, USA). A Bioanalyzer 2100 (Agilent Technologies, Berlin, Germany) was used to monitor the integrity of the RNA samples. RNA-seq was performed with the Illumina-PE150 sequencer (Illumina, San Diego, CA, USA). The NOISeq method was employed to identify differentially expressed genes with a 1.5-fold change between two groups (HCT116-pLV-GREM1 and HCT116-pLV cells) [35]. To clarify the molecular mechanisms underlying the effect of GREM1 on the biological behavior of colorectal cancer cells, gene set enrichment analysis (GSEA) was performed as described previously [36], and was used to explore the enrichment terminology in the pathway of the gene signature Kyoto Encyclopedia of Genes and Genomes (KEGG) related to EMT, UPR, TGFβ, VEGF and PI3K/AKT/mTOR. A heatmap was generated from this data using the R heatmap function. A *p*-value of <0.05 was considered statistically significant.

### 2.4. qRT-PCR

The total RNA of both cell lines was extracted with protocols provided by Trizol (Invitrogen, 15596026). According to the manufacturer’s instructions, cDNA was generated using the PrimeScript RT Reagent Kit with gDNA Eraser (Accurate Biology, AG11706, Hunan, China). The SYBR Green Premix Pro Taq HS qPCR Kit (Accurate Biology, AG11701) was then used to quantify mRNA expression according to the manual instructions. For these experiments, gene-specific and GAPDH primers were employed (Table A1). Data were analyzed using BioRad CFX Manager Software (BioRad, Hercules, CA, USA). Each experiment was prepared in triplicate, and data are represented as mean ± SD of at least three independent experiments. To normalize sample variation, expression of GAPDH was determined as the internal control.

### 2.5. Western Blot Analysis

The total lysate was prepared according to the designated processing procedure. CEB lysis buffer (Invitrogen, FNN0011) was used to harvest the cells. Quantification of proteins was performed according to the manufacturer’s procedure using Pierce’s BCA Protein Assay Reagent Kit (Pierce Biotechnology, 23227, Rockford, IL, USA). Protein bands were visualized using ECL Western blotting substrate (Thermo Fisher, 32132, Rockford, IL, USA) and the ChemiDoc Imager System (Bio-Rad), and analyzed using ImageJ software. Membranes were blocked with bovine serum albumin (BSA, Sigma–Aldrich, A1933, St. Louis, MO, USA) and incubated with one of the following primary antibodies purchased from Cell Signaling Technology (CST, MA, USA): ZEB1 (Cat. #3195, 1:1000), Snail (Cat. #3879, 1:1000), Slug (Cat. #9585, 1:1000), E-cadherin (Cat. #3195, 1:1000), Vimentin (Cat. #5741, 1:1000), ZO-1 (Cat. #8193, 1:1000), β-catenin (Cat. #8480, 1:1000), ATF4 (Cat. #11815, 1:1000), IRE1α (Cat. #3294, 1:1000), ATF6 (Cat. #65880, 1:1000), p-PI3K (Cat. #4228, 1:1000), PI3K (Cat. #4249, 1:1000), p-Akt (Cat. #4060, 1:1000), Akt (Cat. #4691, 1:1000), mTOR (Cat. #2983, 1:1000), p-mTOR (Cat. #5536, 1:1000), Smad1 (Cat. #6944, 1:1000), p-Smad1/Smad5/Smad9 (Cat. #13820, 1:1000), Bip (Cat. #3177, 1:1000) and GAPDH (Cat. #51332, 1:5000). Secondary antibodies were as follows: anti-rabbit antibody (Proteintech, SA00001-2, 1:5000, Wuhan, China) and anti-mouse antibody (Proteintech, SA00001-1, 1:5000). Pre-stained protein molecular weight marker (Thermo Scientific, 26616) was used for protein size calculation. Each experiment was performed in triplicate and the most representative images were selected. The relative band intensity was assessed by densitometric analysis of digitalized autographic images using ImageJ software. To normalize sample variation, expression of GAPDH was determined as the internal control.

### 2.6. Invasion Assay

Cells (1 × 10^5^) were plated in serum-free media in the 24-well “Corning Transwell plate” that was coated with Matrigel (Corning Inc., 354480, NY, USA) for invasion experiments. The lower chamber was filled with DMEM containing 10% FBS. Movements of SW480 and HCT116 cells were measured with five random visual fields quantified by microscopy after a 36 h incubation followed by staining with 0.1% crystal violet. Invasive capacity of cells was assessed with ImageJ software.

### 2.7. Wound Healing Assay

Cells were inoculated into 6-well plates after centrifugation and digestion with 0.25% trypsin. When the cell density reached up to 90% or more, 3 vertical lines were scratched in each well with a 10 µL pipette tip and the floating cells were gently washed away with 1× PBS. Complete medium was added, and a photo was taken of the scratches at 0 h. Three different fields of view were selected for each well for photography. After photography, the medium was changed to a serum-free medium. The healing of the wound was photographed at the same location at the corresponding time of 24 h, 48 h and 72 h after incubation. The experiment was repeated 3 times. To observe the healing effect of drugs on the cell wound, the corresponding drugs (Tunicamycin 0.5 μg/mL, Sigma, T7765; GSK621 30 μM, MCE, HY-100548; CeapinA7 16 μM, Sigma, 2323027-38-7) were added into the serum-free medium. Afterwards, the wound healing was observed under the microscope at 0 h, 24 h, 48 h and 72 h and analyzed using ImageJ software.

### 2.8. Animal Experiments

Animal experimental methods were approved by the Ethical Committee of Sun Yat-sen University (Institutional Animal Care and Use KY-2021-096-02). Six-week-old nude mice were purchased from Zhejiang Vital River Laboratory Animal Technology Co., Ltd. (Hangzhou, China). Mice were kept in specific pathogen-free conditions: 20–24 °C, 12/12 h of dark/light cycle, 60 ± 5% humidity and plastic cage (four mice/cage). HCT116 cells were stably transduced with a lentiviral overexpression vector encoding luciferase according to the manufacturer’s protocol (Shanghai Genechem Co., Ltd., Shanghai, China). Here, HCT116 pLV-GREM1-luc and HCT116 pLV-luc cells (2 × 10^6^) were intravenously injected via mouse tails. GSK621 (30 mg/kg) and CeapinA7 (470 μg/kg) were injected intraperitoneally into the mice once daily, respectively, until day 12. Lung tumor colonization was observed by bioluminescence imaging on the 5th day and 12th day.

### 2.9. Immunohistochemistry

Informed consent was obtained for all specimens collected. The experimental protocol of this study was approved by the Ethics Committee of the Seventh Affiliated Hospital, Sun Yat-sen University. This study abided by the Declaration of Helsinki principles. Immunohistochemical staining for GREM1 (1:100 dilution, Biorbyt, orb10741, Cambridge, UK) was performed on formalin-fixed, paraffin-embedded samples from 55 patients with clinical stage IV CRC. After dewaxing, hydration and antigen repair, the rest of the experimental procedures were performed according to the instructions of the SP Immunohistochemistry Kit (ZSBIO, PV9000, Beijing, China). Finally, after DAB staining, hematoxylin re-staining and neutral resin sealing, the sections were observed under the microscope for the degree of staining.

### 2.10. Statistical Analysis

All experiments were performed in triplicate. GraphPad Prism 8.0 was used for statistical analysis. Student’s *t*-test was used to compare the differences between two groups. Statistical differences of multiple groups were determined by one-way analysis of variance (ANOVA) followed by Tukey’s post hoc tests. The mean ± standard error of the mean (SEM) were used to express the data. A *p*-value of less than 0.05 was considered statistically significant. *, *p* < 0.05.

## 3. Results

### 3.1. The GREM1 Level Is Associated with Poor Prognosis of CRC

To investigate the clinical relevance of GREM1 in CRC, we first stained patient specimens to determine GREM1 expression via IHC. Compared with normal tissues, GREM1 was significantly upregulated in tumor tissues (Figure 1A). Moreover, RNA-seq of the CRC dataset in the Human Protein ATLAS (https://www.proteinatlas.org/ accessed on 30 April 2022) showed that *GREM1* overexpression was significantly associated with poor overall survival (OS). The Kaplan–Meier plots were generated as per the best cutoff value of 5.31, suggesting that a level of *GREM1* expression (FPKM) lower than 5.31 was defined as “Low”, otherwise as “High”(*p* = 0.046; Figure 1B). Thus, by distinguishing high vs. low expression of *GREM1*, we found that the corresponding survival curves between the two groups were separated significantly. These findings suggest that GREM1 may be associated with human CRC progression.

### 3.2. GREM1 Promotes Invasion, Migration and ER Stress of CRC Cells

To evaluate the functional role of GREM1 in CRC cells, we stably overexpressed or knocked down *GREM1* in human CRC SW480 and HCT116 cells using a lentivirus-based system including a *GREM1* gene vector and its control (pLV-GREM1 vs. pLV), and the shRNA targeting *GREM1* and its control (shGREM1 vs. shNC). The protein expression levels of GREM1 were detected by Western blotting that showed that GREM1 protein expression levels were substantially elevated in HCT116 and SW480 cells expressing pLV-GREM1 compared to GREM1 negative control, whereas GREM1 protein expression levels were significantly reduced in shRNA-expressing lentivirus-infected cells (Figure 2A).

Firstly, to identify candidate genes that are sensitive to *GREM1* overexpression, we performed RNA-seq and the subsequent GSEA analysis revealed remarkable cellular phenotypes upon *GREM1* overexpression, such as enriched gene sets associated with EMT activation and unfolded protein response (UPR) (NES = 2.0395, NP = 0.0000; NES = 1.7594, NP = 0.0000, respectively) (Figure 2B). In parallel, the pertinent heatmap derived from the RNA-seq showed that expression of EMT-inducing genes, such as *VEGFA* [37], *CXCL1* [38] and *JUN* [39], were significantly increased, whereas a significant decrease was found in EMT suppressor genes, such as *MCM7* [40], *FBLN1* [41,42] and *DPYSL3* [43]. Interestingly, the majority of UPR downstream genes were significantly upregulated, except for the PERK/ATF4 arm of UPR signaling-associated genes, such as *EIF2S1* (*EIF-2a*) and *EIF2S2* (*EIF-2b*) [44] that were significantly downregulated in HCT116 pLV-GREM1 vs. in HCT116 pLV cells (Figure 2C).

Further, to investigate the changes in migration and invasion capacity upon *GREM1* overexpression or underexpression, we performed wound healing assays which showed that GREM1-overexpressing CRC cells migrated markedly faster than control cells, in keeping with the results of the invasion assays. Meanwhile, knockdown of *GREM1* by GREM1-specific shRNA reduced the invasion and migration of HCT116 and SW480 cells (Figure 2D,E). All these results indicated that *GREM1* promoted migration and invasion of CRC cells. We postulated that these cellular phenotypes might be attributed to EMT and ER stress presented in our aforementioned transriptomic signature. Consistently, expression of canonical EMT factors ZEB1, Vimentin, Snail and ZO-1 were significantly upregulated, and expression of E-cadherin was markedly downregulated at both the mRNA (Figure 2F,G) and the protein levels (Figure 2H) in *GREM1*-overexpressing CRC cells. Previous studies have shown that EMT in a variety of tumor cells is inextricably linked to the activation of ERS, which could be seen in our study that the canonical ER stress chaperone and sensors Bip, ATF6 and IRE1a were upregulated whilst ATF4 was downregulated at the mRNA and protein expression levels in *GREM1*-overexpressing HCT116 and SW480 cells (Figure 2I–K). Taken together, these results indicate that GREM1 promotes EMT and regulates ER stress signaling in CRC cells.

### 3.3. ER Stress Activator Tunicamycin Inhibits Invasion and Migration of CRC

To further determine the sequential role of ER stress and EMT in CRC progression, we assessed the influence of ER stress on the motility of SW480 and HCT116 cells that were treated with a general ER stresser, tunicamycin (TM), and subjected to the wound healing assay. We found that the treatment group showed slower migration, which filled only half of the wound region 48 h after injury (Figure 3A). In keeping, qRT-PCR (Figure 3B,C) and Western blotting (Figure 3D) confirmed that TM treatment reduced the expression of ZEB1 and Vimentin and increased the expression of E-cadherin. As expected, TM treatment significantly increased the expression of ATF4 and Bip in these cells, although ATF6 expression was not affected to a substantial extent (Figure 3D). These results suggest that activated ATF4 signaling may be involved in inhibition of CRC motility and EMT.

### 3.4. GREM1 Promotes CRC Invasion and Migration through Activating ATF6 but Inhibiting the ATF4 Signaling Pathways

To further delineate the explicit arms of the UPR signaling that might be regulated by GREM1, we used our RNA-seq data to perform GSEA that presented negative regulation of PERK (NES = 1.6542, NP = 0.0000) but positive regulation of ATF6 signaling pathways (NES = 1.3553, NP = 0.0000) (Figure 4A). GSK621, a known AMPK activator, also induces PERK phosphorylation and activates the downstream eIF2α/ATF4 signaling pathway [45]. CeapinA7 is a selective blocker of ATF6α signaling in response to ER stress [46,47]. To gain insight into the underlying molecular mechanisms, we used the small molecules GSK621 and CeapinA7 to modulate the activity of these UPR pathways. GSK621 and CeapinA7 caused a significant reduction in cell invasion and migration detected by Transwell assay (Figure 4B) and wound healing assay (Figure 4C,D). Consistent with this result, the EMT-related genes (*ZEB1*, *ZO-1* and *Snail*) were significantly downregulated, and E-cadherin was significantly upregulated in the GSK621 and CeapinA7 treatment group at both the mRNA (Figure 5A–D) and protein levels (Figure 5E,F), respectively, suggesting an EMT suppression role of these two compounds. These results collectively consolidate that ATF6 and ATF4 of the UPR pathways mediate the GREM1-induced CRC invasion and migration in vitro.

### 3.5. GREM1 Modulates ATF4 and ATF6 via Inhibiting BMP and Activating VEGF-VEGFR2-PI3K-AKT Signaling Pathways

GREM1 is known as an antagonist of BMP2 to inhibit TGFβ/BMP signaling [48]. In contrast, activation of the BMP2 signaling pathway promotes the expression of ATF4 [49]. Hence, we postulated that overexpression of GREM1 in CRC cell lines would inhibit ATF4 expression by canonically suppressing the BMP2 signaling pathway. On the other hand, GREM1 was reported to non-canonically bind VEGFR2 and activate various intracellular effectors downstream [50]. The PI3K-AKT-mTOR axis, a classical downstream pathway of VEGF [51], could promote ATF6 expression [52]. We thus hypothesized that GREM1 might activate VEGFR2 and its downstream PI3K-AKT-mTOR axis to elevate ATF6 expression. In support, GSEA of our in-house RNA-seq dataset revealed the enrichment of the PI3K/AKT/mTOR, VEGF-VEGFR2 and TGFβ signaling pathways in HCT116 pLV-GREM1 cells (Figure 6A,B). In keeping, the heatmaps derived from the RNA-seq displayed that genes associated with VEGF/VEGFR2 and PI3K/AKT/mTOR signaling pathways, such as *MAP2K1* (*MEK1*), *MAPK14* (*p38*), *CDC42*, [51,53], *FOXO4* [54], *LDLR* [55] and *HPRT1* [56], were upregulated in response to GREM1 overexpression, whilst genes of the TGFβ signaling pathway were suppressed, such as *ID1*, *ID2* and *TGFB1* [57] (Figure 6C). Next, to validate these findings, we assessed the changes in protein expression levels of the key downstream molecules involved in these pathways by overexpression or knockdown of *GREM1* in CRC cells. The results showed that overexpression of GREM1 promoted the activation of the PI3K/AKT/mTOR axis, downstream of the VEGF-VEGFR2 pathway, but impeded BMP-Smad1/5/9 signaling (Figure 6D,E). On the contrary, knockdown of *GREM1* resulted in declined expression of phosphorylated PI3K, phosphorylated AKT and phosphorylated mTOR but in elevated expression of phosphorylated Smad1/5/9 (Figure 6D,E).

### 3.6. Effects of GREM1, GSK621 and CeapinA7 on CRC Metastasis In Vivo

To further evaluate our findings in vivo, luciferase-encoding lentiviral overexpression vectors (luc) were successfully transduced into HCT116 cells. HCT116-luc cells (pLV-luc or pLV-GREM1-luc) were injected into the tail veins of nude mice and tumor progression was then monitored with bioluminescence imaging. *GREM1* overexpression significantly increased tumor metastasis to the lung. By contrast, treatments by GSK621 which could activate ATF4 or by CeapinA7, an ATF6 inhibitor, significantly suppressed tumor metastasis rates (Figure 7A–C). These results further corroborate that GREM1 could promote CRC metastasis through activating the ATF6 pathway but inhibiting the ATF4 pathway (Figure 8).

## 4. Discussion

GREM1, a BMP antagonist, is thought to play a role in organogenesis, body patterning and tissue differentiation [48,58,59]. Canonically, Gremlin1 directly binds and inhibits BMP2 and BMP4 [48]. Furthermore, GREM1 also participates in the EMT through regulating the STAT3-MMP3 signaling, in addition to the BMP signaling pathway [60]. Non-canonically, previous studies have demonstrated that GREM1 binds to VEGFR2 and promotes angiogenesis [50]. The effect of GREM1 on the prognosis of malignant tumors remains elusive. Some studies have linked high GREM1 expression to poor prognosis in several malignancies, including ER-negative breast cancer [16], cervical cancer [61], ex-trahepatic cholangiocarcinoma [62], basal cell carcinoma [63] and renal cell carcinoma [64]. In CRC, high GREM1 protein expression has been associated with low tumor stage and extended survival [20]. However, our study showed that high GREM1 expression was significantly correlated with CRC progression.

GREM1 was found to be overexpressed in an array of common neoplasms of cervix [65], ovary [66], lung [67], stomach [68], breast [15] and kidney [64], and to interact with YWHAH protein to exert a pro-carcinogenic effect [69]. GREM1 was also overexpressed in malignant mesothelioma [70], pancreatic neuroendocrine tumors [71] and hepatocellular carcinoma associated with hepatitis C infection [72]. It was reported that high GREM1 expression in gliomas and cervical cancers played an important role in maintaining the stemness of cancer stem cells [65,73]. In our study, we found that GREM1 was highly expressed in stage IV CRC tissues and was strongly associated with poor prognosis. We also found that high GREM1 expression promoted EMT in CRC cells. There are sporadic reports of detailed insights into the molecular mechanisms of GREM1-mediated tumor-igenesis. GREM1 is associated with AKT/mTOR signaling in lung malignant mesothelioma [18]. GREM1 in breast cancer cells promote ERK activation [74]. GREM1 in Caco2 colon cancer cells inhibits differentiation by suppressing p21/CKDN1A expression [75]. Several reports have identified that GREM1 amplifies TGFβ1 signaling to drive EMT in CRC [76] and esophageal squamous cell carcinoma [77]. Our study illustrated that overexpression of *GREM1* in CRC cells modulated ER stress, through which GREM1 promoted EMT via activation of the ATF6 and inhibition of the ATF4 pathways.

Protein handling, modification and folding in the endoplasmic reticulum (ER) are tightly regulated processes that determine cell function, fate and survival [30]. Studies have shown that the signaling molecules of ER stress affected tumor metastasis by reg-ulating EMT-transcription factors [78,79,80]. As Tunicamycin (TM) is widely used as an ER stress inducer in experimental settings due to its inhibitory effect on N-linked glycosylation [81,82], we used TM to treat CRC cells to observe the effect of ER stress on EMT. The UPR induced by ER stress is one of the most important mechanisms regulating cellular adaptation to an adverse microenvironment [83]. TM treatment in CRC cells profoundly suppressed the EMT, which is largely due to activation of the PERK/ATF4 arm of UPR. Since we did not observe a drastic change of ATF6 thereafter, it promoted us to investigate further how these two signature pathways orchestrate the EMT. We used small molecules to modulate the activity of these UPR pathways, namely GSK621 to induce PERK-eIF2α-ATF4 pathway [84] and CeapinA7 to block the activation of ATF6α [85]. By those approaches, our study demonstrated that GREM1 could regulate ER stress and EMT by activating ATF6 and inhibiting ATF4 signaling pathways. To further explore the intrinsic molecular mechanism underlying GREM1 regulation of ATF4 and ATF6 signaling, we conducted transcriptomic analysis and integrated it with previous studies. We observed that GREM1 inhibited TGFβ/BMP signaling, which was possibly through antagonizing BMP2 [48] and thereby suppressing ATF4 expression [49]. It was reported that GREM1 integrates with VEGFR2 to activate the PI3K/AKT/mTOR signaling pathway thereby promoting ATF6 expression [51,52]. We found that GREM1 activated PI3K/AKT/mTOR, a classical downstream of the VEGF-VEGFR2 signaling pathway that was likely to promote ATF6 expression. Taken together, these results implied that GREM1 induced EMT in CRC cells by inhibiting ATF4 expression and promoting ATF6 expression through the regulation of TGFβ/BMP and VEGF/VEGFR2, respectively. Detailed mechanisms as to how ATF6 and ATF4 divergently regulate CRC cell metastasis are worth further investigation in the future.

In conclusion, our study demonstrated that GREM1 was associated with CRC pro-gression, and it may play an important pro-metastatic role. This could be mediated, at least in part, by the divergent regulation of the ATF6 and ATF4 of the UPR pathways through activation of PI3K/AKT/mTOR and antagonization of BMP2 signaling pathways.

## Figures and Tables

**Figure 1 cells-11-02136-f001:**
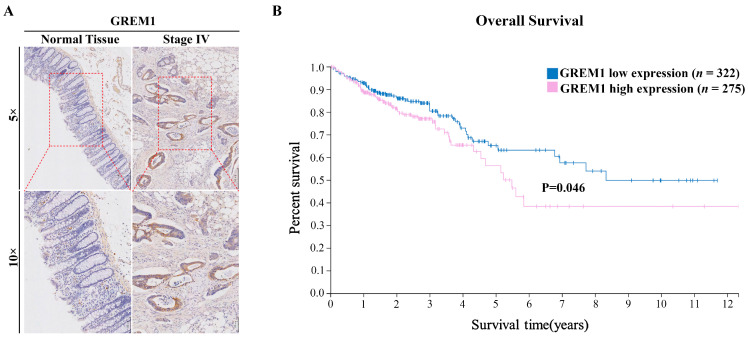
**GREM1 expression is correlated with poor survival in CRC patients.** (**A**) Representative images of GREM1 in para-carcinoma and carcinoma tissues of patients with stage IV CRC. Scale bar, 250 µm. (**B**) Kaplan–Meier survival analysis of 597 colon cancer patients stratified by *GREM1* expression levels. Kaplan–Meier survival analysis and *GREM1* mRNA expression according to FPKM value 5.31 as a cutoff value in the whole cohort. *p* = 0.046 by log-rank (Mantel–Cox) test.

**Figure 2 cells-11-02136-f002:**
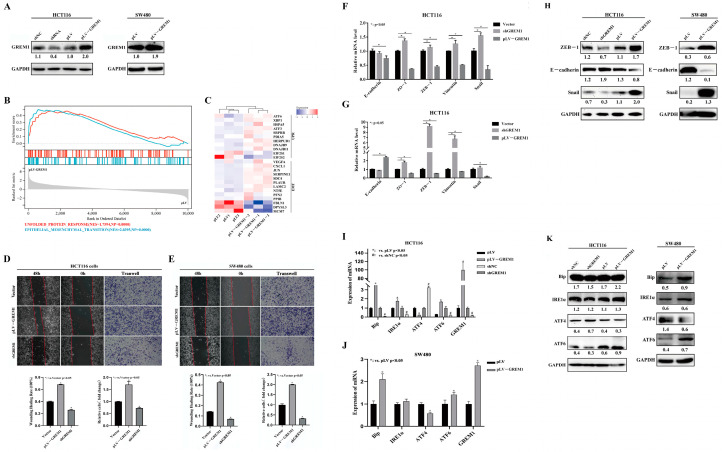
**GREM1 expression promotes cell invasion and migration of HCT116 and SW480 cells and ER stress.** (**A**) Overexpression or underexpression of *GREM1* was detected by Western blotting. (**B**) GSEA was performed using our RNA-seq dataset to identify sets of hallmark genes that were positively associated with GREM1 expression. Enrichment plots showed EMT (NES = 2.0395, NP = 0.0000) and UPR (NES = 1.7594, NP = 0.0000). (**C**) The heatmap showed signature genes involving EMT and UPR. (**D**,**E**) Wound healing test and Boyden chambers with Matrigel revealed that *GREM1* overexpression promoted HCT116 and SW480 cell migration and invasion. (**F**–**H**) Cells were analyzed after harvesting, for protein and mRNA expression. qRT-PCR and Western blotting analysis showed expression of mRNA and proteins involved in EMT. (**I**–**K**) qRT-PCR and Western blotting results showed the expression of proteins and mRNA that were involved in UPR and ER stress.

**Figure 3 cells-11-02136-f003:**
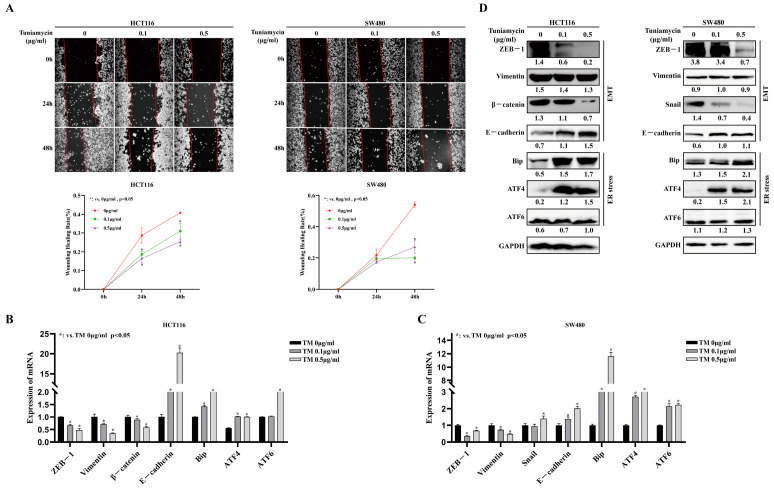
**Tunicamycin inhibits epithelial-to-mesenchymal transition in human CRC cell lines.** SW480 and HCT116 cells were exposed to tunicamycin (0.1 μg/mL, 0.5 μg/mL). (**A**) Wound healing test revealed that tunicamycin inhibited HCT116 and SW480 cell migration. (**B**–**D**) SW480 and HCT116 cells were exposed to tunicamycin (0.1 μg/mL, 0.5 μg/mL) for 6 h, and then analyzed for protein and mRNA expression. qRT-PCR and Western blotting results showed the expression of proteins and mRNA involved in EMT.

**Figure 4 cells-11-02136-f004:**
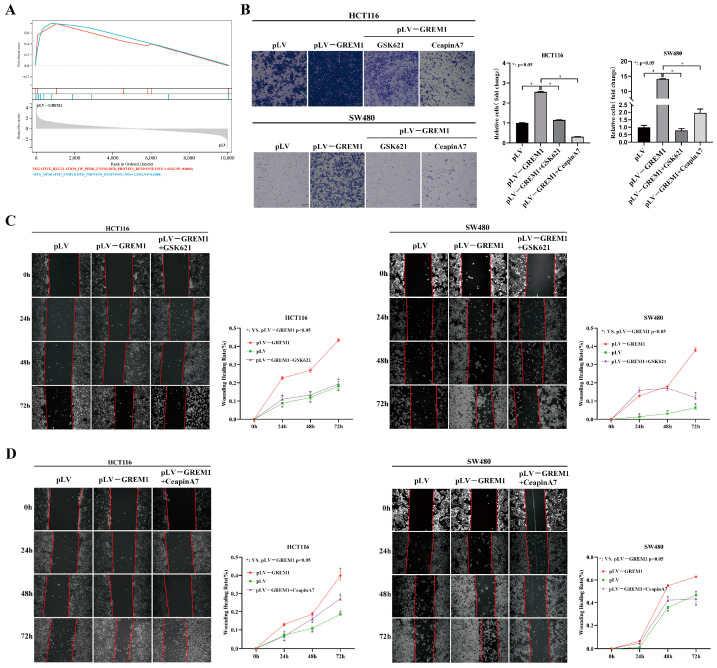
**CeapinA7 and GSK621 inhibit GREM1-induced CRC cell****line invasion and migration**. (**A**) The GSEA was performed using an in-house RNA-seq dataset to identify gene sets in the KEGG collections that have positive correlations with GREM1 expression. The negative regulating PERK-mediated UPR (NES = 1.6542 NP = 0.0000) and ATF6-mediated UPR signaling pathway (NES = 1.3553, NP = 0.0000) with GREM1 overexpression in HCT116 cells. (**B**) HCT116 pLV-GREM1 and SW480 pLV-GREM1 cells were seeded into Matrigel-coated inserts with medium containing CeapinA7 (16 μM) or GSK621 (30 μM) and incubated for 36 h, followed by the invasion assay. (**C**,**D**) Wound healing test revealed that CeapinA7 and GSK621 inhibited cell migration.

**Figure 5 cells-11-02136-f005:**
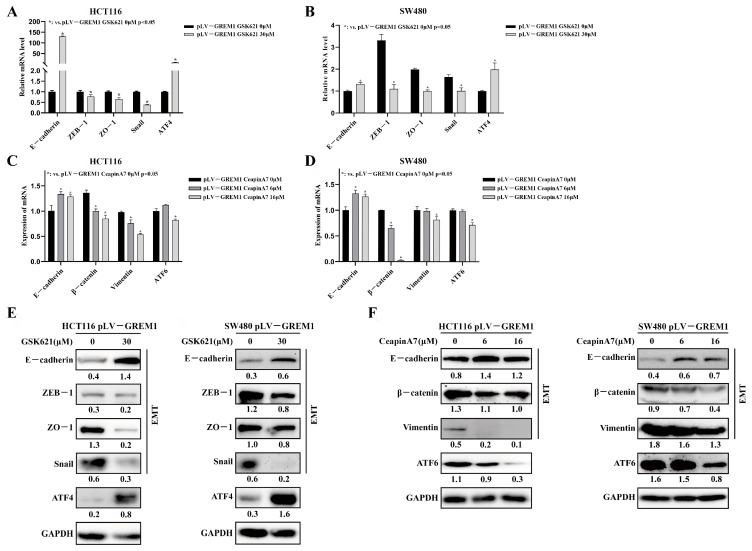
**CeapinA7 and GSK621 reverse EMT in *GREM1*-overexpressing CRC cell lines.** HCT116 pLV-GREM1 and SW480 pLV-GREM1 cells were exposed to CeapinA7 (6 μM,16 μM) or GSK621 (30 μM) for 24 h, respectively. After harvesting, the cells were analyzed for protein and mRNA expression. (**A**–**D**) qRT-PCR analysis revealed the expression of mRNA involved in EMT. (**E**,**F**) Western blotting analysis revealed the expression of proteins involved in the EMT.

**Figure 6 cells-11-02136-f006:**
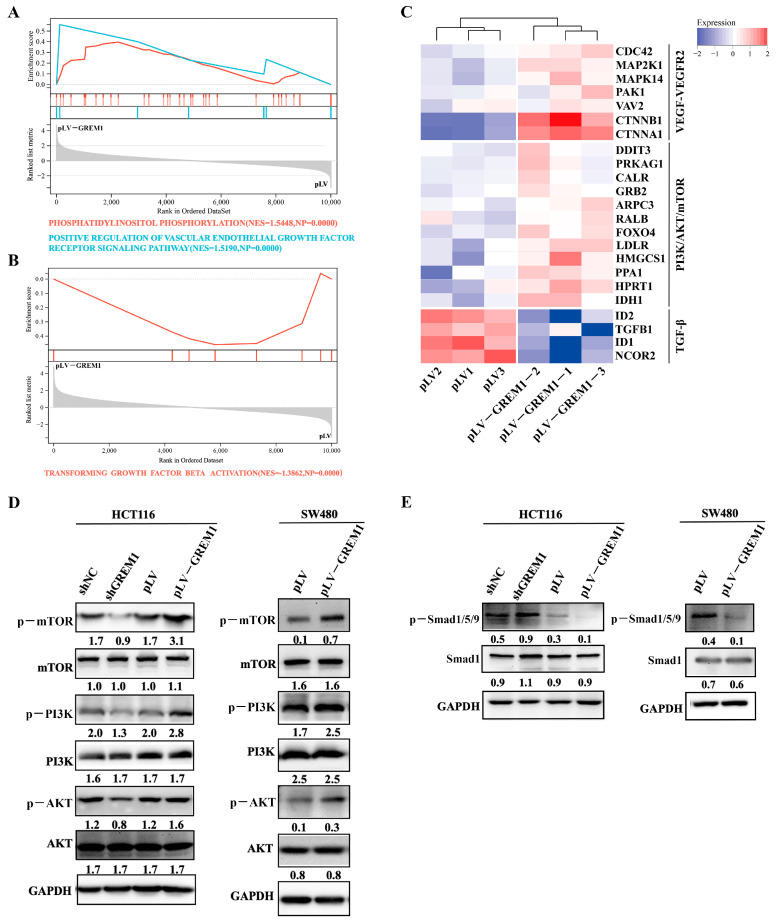
**GREM1 negatively regulates ATF4 and but positively regulates ATF6 via suppressing BMP and activating the VEGF-PI3K-AKT signaling pathways.** (**A**,**B**) GSEA of RNA-seq data showed enriched gene sets associated with VEGF/VEGFR2, PI3K/Akt/mTOR and TGFβ signaling pathways annotated in the KEGG by comparing *GREM1* overexpression versus negative control CRC cells. (**C**) Heatmap showing gene expression differences in the pertinent signaling pathways. (**D**,**E**) Western blotting analysis revealed the expression of proteins involved in the pertinent signaling pathways.

**Figure 7 cells-11-02136-f007:**
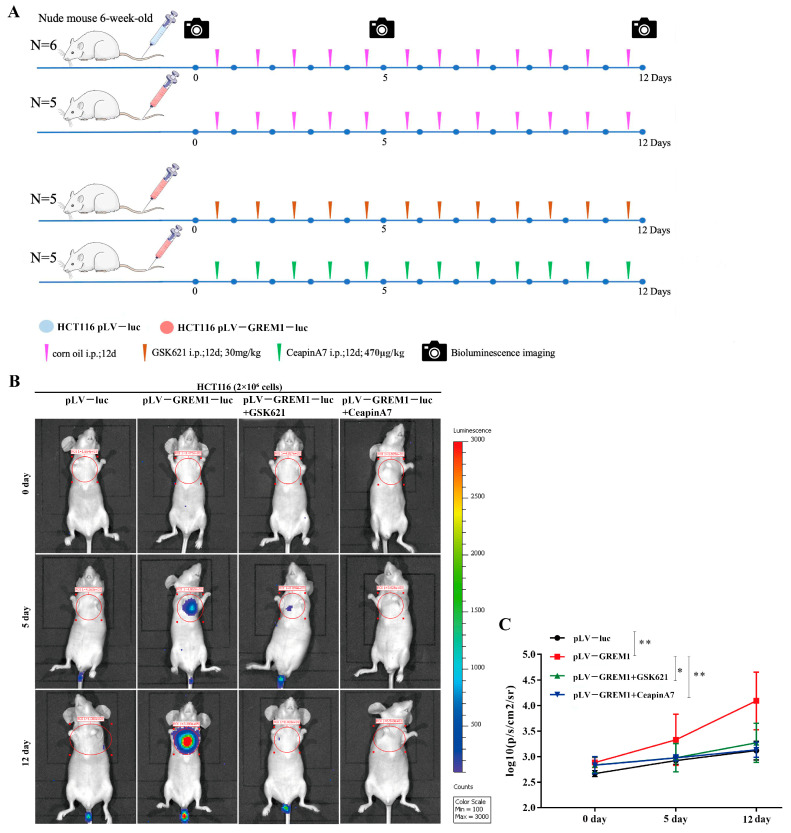
**Anti-tumor activity of CeapinA7 and GSK621 in vivo.** (**A**) Animal experimental grouping and plans. (**B**) After the injection of different cells (2 × 10 ^6^ cells/nude mouse) via tail vein of nude mice, the distribution of CRC cells in these mice was detected via a bioluminescence imaging (BLI) system. The scale to the right of the BLI images describes the color map for the luminescent signal. (**C**) Fluorescence intensity of lung metastasis areas in different groups of the CRC cells. *n* = 21. * *p* < 0.05, ** *p* < 0.01.

**Figure 8 cells-11-02136-f008:**
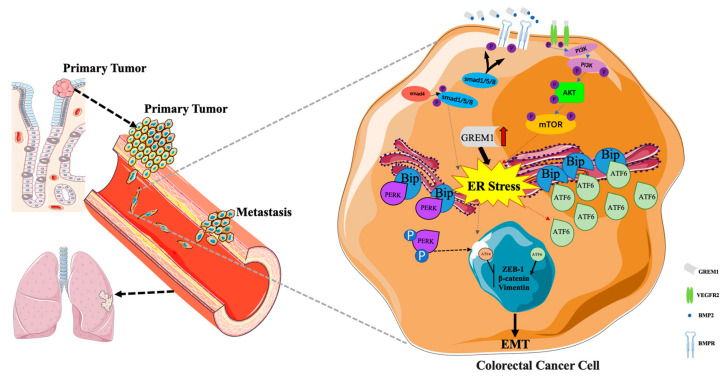
Mechanistic diagram of the role of GREM1 in ER stress, EMT and metastasis in CRC cells.

## Data Availability

All data described are in this article.

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
