# Peer review of "Gremlin-1 Promotes Colorectal Cancer Cell Metastasis by Activating ATF6 and Inhibiting ATF4 Pathways"

_cells, 2022, doi:10.3390/cells11142136_

Round 1

Reviewer 1 Report

The original article written by Dr. Li and colleagues entitled “Gremlin1 promotes colorectal cancer cell metastasis by activating ATF6 and inhibiting ATF4 pathways” describes the effect of Gremlin1 in colorectal cancer metastasization by hypothesizing the implication of endoplasmic reticulum’s stress due to the overexpression of gremlin.

This paper is well wright and clear, however, seems to be very little depth. Gremlin1 modulates a great number of processes both as a BMPs antagonist both as a VEGFR2 ligand. It is also well known for its pro-inflammatory, pro-fibrotic, and EMT-inducer protein.

Too many questions remain unanswered.

Is the regulation of colorectal cancer cell behavior autocrine or paracrine?

Which receptor mechanics can be involved?

Is the effect on ATF4/ATF6 direct or indirect?

Since no mechanism of action and regulation is proposed, I do not think that this work can be accepted in the present form.

Reviewer 2 Report

The paper named “Gremlin-1 promotes colorectal cancer cell metastasis by activating ATF6 and inhibiting ATF4 pathways” demonstrated that GREM1 is an invasion-promoting factor via regulating expression of ATF6 and ATF4 in colorectal cancer cells. Seeing these results authors suggested that GREM1 may be a potential pharmacological target for colorectal cancer treatment.

The results are clearly expressed although some points need to be clarified.

  1. In Materials and Method section, in the part of Western blot analysis, information about the antibodies and the concentration used were missed.
  2. In Materials and Method section, in the part of would healing author say that “observe the healing effect of drugs on the cell wound, the corresponding drugs were added into the serum-free medium” line 126-127 what drug was used? What concentrations were used?
  3. In Materials and Method section, in the part of Inmunohistochemistry, information about the antibodies and the concentration used were missed.
  4. Nothing about the overexpresion of GREM1 used in the study were explained in Materials and Method section.
  5. In paragraph between line 175-176 author explain results that are not shown.
  6. In the same way as question 4 how author make the knockdown of GREM1 using a shRNAs? What type of sh is used? how is it introduced into cells?
  7. In figure 1F, why author do not analyze ATF4? This data is interesting in these different conditions.
  8. In figure 1, what means KDNC or OENC? This information must be added to figure caption to help to understand the results.
  9. In point 3.5 authors only demonstrate that ATF6 was activated and ATF4 was inhibited but not that those were implicate in the process. Has author perform some ATF6 inhibition? or ATF4 sobreepression to corroborate the date?

Reviewer 3 Report

Li et al. investigated the functional roles of Gremlin-1 in promoting colorectal cancer metastasis using in vitro cell line and in vivo mouse models. Although the findings are interesting, the conclusions are not exactly consistent with the evidence provided. The following points need to be considered by the authors.

  1. In Fig. 1, the authors need to clarify the criteria of low and high expression of Gremlin-1. What the level of Gremlin-1 expression was defined as low, or high?
  2. In Fig. 2, the authors need to provide evidence that Gremlin-1 protein expression was upregulated or downregulated in GREM1-OE and GREM1-sh3 cell lines, respectively. In panel F, ATF4 mRNA level should also be checked in different HCT116 cell lines. In panel H, the protein level of ATF4 is upregulated in GREM1-OE HCT116 cell line, but downregulated in GREM1-sh3 HCT116 cell line, which disagrees with the description and conclusion in the results (line 190 and 191).
  3. In Fig. 3, it looks like tunicamycin treatment also increases ATF6 expression. If so, why tunicamycin treatment can inhibit invasion and migration of CRC if ATF6 has a positive role in this process? In panel E (left), the authors showed the relative protein level of GAPDH in right line is 1.0, and in the left and middle lines are 0.6 and 0.7, respectively. However, the right line of GAPDH should have the lowest expression level based on the image intensity. So, the western blot needs to be re-analyzed.
  4. GSK621 is an AMPK activator which might also induce UPR activation, while CeapinA7 is an inhibitor of ATF6 which could inhibit UPR pathways. However, the authors got the same or similar results when treated GREM1-OE cell lines with these two different drugs. Based on Figure 4 -6, the authors cannot make the conclusion that ATF6 and ATF4 of the UPR pathways mediate the GREM1-induced CRC metastasis in vitro or in vivo (line 214 and 215). Neither ATF6 nor ATF4 was directly functionally analyzed in those experiments.

Round 2

Reviewer 1 Report

I have no further questions for the revised manuscript. 

Reviewer 3 Report

The resolution of the figures is too low to be appreciated.

Please provide the figures with higher resolution.

This manuscript is a resubmission of an earlier submission. The following is a list of the peer review reports and author responses from that submission.